# Medical Faculty’s and Students’ Perceptions toward Pediatric Electronic OSCE during the COVID-19 Pandemic in Saudi Arabia

**DOI:** 10.3390/healthcare9080950

**Published:** 2021-07-28

**Authors:** Lana A. Shaiba, Mahdi A. Alnamnakani, Mohamad-Hani Temsah, Nurah Alamro, Fahad Alsohime, Abdulkarim Alrabiaah, Shahad N. Alanazi, Khalid Alhasan, Adi Alherbish, Khalid F. Mobaireek, Fahad A. Bashiri, Yazed AlRuthia

**Affiliations:** 1College of Medicine, King Saud University, Riyadh 11362, Saudi Arabia; lshaiba@ksu.edu.sa (L.A.S.); malnamnakani@ksu.edu.sa (M.A.A.); nmalamro@ksu.edu.sa (N.A.); fAlsohime@ksu.edu.sa (F.A.); Alrabiaah@ksu.edu.sa (A.A.); Shahad.n.f.a@gmail.com (S.N.A.); kalhasan@ksu.edu.sa (K.A.); aalherbish@KSU.EDU.SA (A.A.); KHALIDFM1@yahoo.com (K.F.M.); fbashiri@ksu.edu.sa (F.A.B.); 2Neonatal Intensive Care Unit, Department of Pediatrics, College of Medicine, King Saud University Medical City, King Saud University, Riyadh 11362, Saudi Arabia; 3General Pediatric Unit, Department of Pediatrics, College of Medicine, King Saud University Medical City, King Saud University, Riyadh 11362, Saudi Arabia; 4Pediatric Intensive Care Unit, Pediatric Department, College of Medicine, King Saud University Medical City, King Saud University, Riyadh 11362, Saudi Arabia; 5Undergraduate Committee, Pediatric Department, King Saud University, Riyadh 11362, Saudi Arabia; 6Department of Family and Community Medicine, College of Medicine, King Saud University Medical City, King Saud University, Riyadh 11362, Saudi Arabia; 7Pediatric Infectious Diseases Unit, Pediatric Department, College of Medicine, King Saud University Medical City, King Saud University, Riyadh 11362, Saudi Arabia; 8Pediatric Nephrology Unit, Pediatric Department, King Saud University Medical City, King Saud University, Riyadh 11362, Saudi Arabia; 9Pulmonary Medicine Unit, Pediatric Department, King Saud University Medical City, King Saud University, Riyadh 11362, Saudi Arabia; 10Pediatric Neurology Unit, Pediatric Department, College of Medicine, King Saud University Medical City, King Saud University, Riyadh 11362, Saudi Arabia; 11Department of Clinical Pharmacy, College of Pharmacy, King Saud University, Riyadh 11451, Saudi Arabia; yazeed@ksu.edu.sa

**Keywords:** final year medical students, COVID-19, pandemic, distance learning, assessment, educational, pediatric, OSCE

## Abstract

Background: The educational process in different medical schools has been negatively affected by the COVID-19 pandemic worldwide. As a part of the Saudi government’s attempts to contain the spread of the virus, schools’ and universities’ educational activities and face-to-face lectures have been modified to virtual classrooms. The purpose of this study was to explore the perceptions of the faculty and the students of an electronic objective structured clinical examination (E-OSCE) activity that took place during the COVID-19 pandemic in the oldest medical school in Saudi Arabia. Methods: An e-OSCE style examination was designed for the final-year medical students by the pediatrics department, College of Medicine at King Saud University in Riyadh, Saudi Arabia. The examination was administered by Zoom™ video conferencing where both students and faculty participated through their laptop or desktop computers. In order to explore the students’ and the faculty’s perceptions about this experience, a newly designed 13-item online questionnaire was administered at the end of the e-OSCE. Results: Out of 136 participants (23 faculty and 112 students), 73 respondents (e.g., 54% response rate) filled out the questionnaire. Most of the respondents (69.8%) were very comfortable with this new virtual experience. Most participants (53.4%) preferred the e-OSCE compared to the classic face-to-face clinical OSCE during the pandemic. Regarding the e-OSCE assessment student tool, 46.6% reported that it is similar to the classic face-to-face OSCE; however, 38.4% felt it was worse. Conclusions: The e-OSCE can be a very effective alternative to the classic face-to-face OSCE due to the current circumstances that still pose a significant risk of infection transmission. Future studies should examine different virtual strategies to ensure effective OSCE delivery from the perspective of both faculty and students.

## 1. Introduction

The educational process throughout the different undergraduate and graduate medical institutes has been immensely disrupted due to the concern of COVID-19 infection transmission as well as the precautionary lockdown and other preventive actions that have been taken to contain the pandemic worldwide [1]. Therefore, many medical schools across the world have adopted and implemented the electronic Objective Structured Clinical Examination (e-OSCE) as a tool to evaluate their medical students with great success [1,2,3,4]. In Saudi Arabia, as part of the government’s attempts to contain the spread of the virus, the face-to-face activities of all educational institutions were converted to the virtual classroom to promote social distancing. Many studies suggested that 90% of the teachers were motivated to implement social education despite having diverse students with different cultural backgrounds [5,6].

The school of medicine at King Saud University offers a 6-year Bachelor of Medicine and Bachelor of Surgery (MBBS) program where the first year includes basic sciences followed by three years of basic clinical courses (e.g., anatomy, physiology, pharmacology, biochemistry), and the last two years are mainly applied clinical courses (e.g., surgery, internal medicine, otorhinolaryngology, ophthalmology, family medicine, and pediatrics). The pediatrics course consists of two major parts (theoretical and clinical). This course is divided into three parts and is delivered throughout the academic year. However, the course delivery has been disrupted by the COVID-19 pandemic, especially for students in their last two years. Therefore, the different online assessment tools, such as the e-OSCE, have been commonly utilized during this pandemic to ensure the continuity of education for medical students and to assess the quality of the educational outcomes [4]. In this study, our objective was to examine the perceptions of students and faculty on the implementation of e-OSCE during COVID-19 lockdown in King Saud University in Riyadh, Saudi Arabia. The perceptions of the fifth-year medical students as well as their faculty members who participated in assessing their performance in the pediatric e-OSCE delivered through Zoom™ teleconferencing platform (Zoom Video Communication, Inc., San Jose, CA, USA) at King Saud University in Riyadh, Saudi Arabia, were explored using a newly developed online-based questionnaire. Furthermore, the steps taken to prepare and conduct this new virtual experiment in the oldest medical school in Saudi Arabia are described.

## 2. Methods

### 2.1. Study Design

This was an online questionnaire-based cross-sectional study that followed a pediatric e-OSCE for the fifth-year medical students to explore the students’ and faculty perceptions toward this new experience during this unprecedented pandemic time at King Saud University in Riyadh, Saudi Arabia. The students underwent an intensive overview and teaching of common pediatrics case presentations two months prior to the e-OSCE over a two-week period. Students were divided into small groups for case discussion, and each group was assigned two faculty members to facilitate the case discussion and observe the performance of each group using a standardized assessment scale. The students were given 120 min (2 h) to discuss the cases, and discussions were held on Zoom™ breakout rooms, given the students’ familiarity with this application. This e-OSCE was conducted as a part of the clinical evaluation for the last course in the fifth-year of medical school. The clinical cases were carefully chosen and included different parts to assess the history and physical examination skills as well as the skills in interpreting radiological and laboratory results. These cases were predesigned and prepared by the undergraduate committee in the department of pediatrics, but did not include real patients. The students were informed about the exam through an email that was sent by the director of the exam committee and included a clear description of the examination process as well as the date/time of the exam and the registration process.

### 2.2. Electronic-OSCE Procedures

The following were steps taken to ensure smooth conduct of the examination:Zoom™ breakout rooms were created by the examination committee.Students arrived 15 min before the start of the exam and were admitted to waiting virtual breakout rooms.The students and the examiners were asked to keep their cameras on throughout the exam.The students were admitted to their assigned virtual breakout rooms where their examiners were waiting for them there (e.g., two examiners in each breakout room).The examiners then presented students with three different clinical scenarios, which included patient history taking, an emergency case, and a chronic pediatric problem.The students were then given eight minutes to answer all post-encounter prompts.While one of the examiners was observing and grading the students, the other examiner acted as a standardized patient as needed based on the case scenario.At the end of the encounter (24 min in total), a five-minute break was given to allow timely admission of the next group of students.A 15-min break between each student and the next was taken to complete the checklist and mark the students by each examiner separately for the three stations.The case scenarios were changed for each group with a total of 18 scenarios.

There were 18 different grading subcommittees with two pediatric consultants (e.g., examiners) in each, which brought the total number of pediatric consultants to 36. The e-OSCE lasted about 2 h per day and was completed in two days.

The objectives of the OSCE stations were to test the student’s ability to take a comprehensive history; including the different unique components in common pediatric cases. The e-OSCE stations were similar in themes and structure to the traditional OSCE that were conducted face-to-face before the COVID-19 crisis.

The general themes of the stations included: taking developmental history and estimating the child’s developmental age, the asthma station with questions in acute management and long-term management, the bronchiolitis station. Also included was a febrile seizure station, and the questions included history taking, differential diagnosis, and distinguishing between febrile seizures and other types of seizures. In addition, a station included interpretation of different patterns of the growth chart. A station was targeted at identifying different equipment in pediatrics and the ability to describe the indications, contradictions, and adverse events.

Furthermore, case stations including abdominal pain and abdominal mass, as well as neonatology or critical care station, were also encompassed. Each station’s allocated time was 8 min per station. Each student had five stations to go through. Each station of the OSCE had 3–4 questions with 2 min of time allocated to answer each question.

### 2.3. Faculty’s and Students’ Perceptions of e-OSCE

In order to evaluate the faculty’s and students’ perceptions of the pediatric e-OSCE during the COVID-19 pandemic, a-13 item online questionnaire was developed by the undergraduate programs committee at the department of pediatrics. This questionnaire consisted of two parts. The first part consists of four questions about demographic characteristics (e.g., age, gender, title) and previous exposure to a teleconferencing experience (e.g., Zoom™, Microsoft teams, webinars); and the second part consists of nine questions about the likelihood to recommend this experience to other students and faculty members, how comfortable this experience was, personal preference to face-to-face or virtual OSCE, how the virtual OSCE affected the quality of student’s assessment, the personal rating of this experience, whether this experience resulted in lower stress and anxiety in comparison to the classic face-to-face experience, and whether it should be incorporated in the future assessment of medical students especially after the end of this pandemic, the different positive aspects of this experience, and the obstacles faced during this experience (Appendix A). The face and content validity of the questionnaire were checked by five faculty members, and the reliability was checked using the Cronbach’s alpha method. All students (103 students) and faculty members (36 faculty) who participated in this e-OSCE were invited to participate in this survey directly after the end of the assessment. Those who accepted the invitation were asked to consent to participate before filling out the questionnaire. No personal identifiers were collected, and the study adhered to the ethical principles of the Helsinki declaration.

### 2.4. Statistical Analysis

The mean and standard deviation were used to describe continuous variables and frequencies and percentages for the categorical variables. The median values were quoted for the bivariate comparisons with the non-parametric comparison methods. The multiple response dichotomy analysis was used to describe the questions measured with multiple option selections. The chi-squared test of association was used to assess the correlations between categorically measured variables. The non-parametric Mann–Whitney U test was used to compare people’s perceptions of the e-OSCE across binary categorical variable levels. Alpha significance level was considered at 0.050 level. All statistical analyses were performed using SPSS 21 (IBM Corp., Armonk, NY, USA).

## 3. Results

Out of 136 participants in the pediatric e-OSCE who were invited to participate in the questionnaire upon the completion of the examination, 73 individuals (23 faculty and 50 students) filled out the questionnaire (54% response rate). The majority of the participants were female (86%) and aged 30 years and younger (68.5%). Almost all participants (98.6%) had previous experience with the Zoom™ videoconferencing platform (Table 1). The questionnaire showed good internal consistency (e.g., Cronbach’s alpha = 0.955).

Most of the participants (69.8%) felt very comfortable or extremely comfortable during their participation in the pediatric e-OSCE. Moreover, most participants preferred the e-OSCE over the classic face-to-face OSCE (53.4%) or did not prefer classic face-to-face (15.1%). In addition, most of the participants (74%) reported that the participation in e-OSCE had either reduced their level of stress or anxiety or did not have any effect in comparison to the classic face-to-face OSCE. However, less than 40% of the participants recommended incorporating e-OSCE in the curriculum after the end of the COVID-19 pandemic, and 74% did not recommend the use of remote students’ assessment in the future (Table 2). When asked on a 1–10 Likert scale about how likely is it that they would recommend such virtual assessment (e-OSCE) to a colleague, their mean (SD) 7.75 (1.63) indicated that most of them would recommend it.

The participants perceived the following as the top three positive contributors to e-OSCE: clear instructions provided, the organizers’ prompt communications, and the free use of the zoom application (Figure 1). Conversely, the slow internet connectivity and the vague instructions, as well as other negative aspects that were not clearly specified, were the most commonly reported negative aspects of the e-OSCE, as shown in Table 3.

To better understand why participants had perceived different quality of e-OSCE assessment, a bivariate analysis was conducted (Table 4). The resulted findings showed that the student’s sex did not correlate significantly on their perception of the quality of e-OSCE assessment, but ages older than thirty years perceived the quality of e-OSCE assessment significantly worse (*p* = 0.030). Also, medical students perceived the quality of e-OSCE assessment significantly worse than teachers (*p* = 0.030) according to the chi-squared test of independence. The non-parametric Mann–Whitney U test showed that physicians who perceived worse e-OSCE quality were significantly less willing to recommend this virtual assessment (Median = 6) as compared to those who perceived it as better or similar to the face-face assessment (Median = 8) on average (U = 350.5, df = 73, *p* = 0.003).

Another non-parametric Mann–Whitney U test showed that the students who perceived worse e-OSCE quality had perceived significantly lower comfort (Median = 3.4) than those who had perceived the e-OSCE equivalent or better with respect to the quality of assessment compared to the face-face methods (U = 418, df = 73, *p* = 0.010). Even so, participants who preferred the face-face OSCE assessment were found to be significantly more predictable of perceiving the e-OSCE as being significantly worse than face-face methods (*p* = 0.022). People’s perception of anxiety during the e-OSCE did not correlate significantly with their perceived quality of the virtual assessment (*p* = 0.093). However, those who perceived the e-OSCE as worse had measured significantly lower willingness to incorporate virtual assessments in future courses (median = 2) compared to people who were satisfied with the quality of the e-OSCE method (median = 4) according to a Mann–Whitney U test (*p* < 0.001).

## 4. Discussion

The current COVID-19 pandemic has emerged as the greatest threat to global educational and healthcare systems and imposed an enormous challenge. Over the last year, virtual education played a vital role in mitigating the impact of this pandemic on the educational process by providing an interactive communication platform. These platforms, such as Microsoft teams^®^, Zoom™, Skype^®^, and Cisco Web™, have been increasingly adopted by all the medical schools across the country to overcome the raised challenges during the COVID-19 pandemic and try to maintain/rescue the educational process through distance learning [7,8]. As the pandemic persisted, another challenge was raised by the end of the academic year of how to safely but reliably assess the performance of the medical students. To ensure the safety of staff and students during this period and maintain physical distancing, the pediatric department implemented an online exam using an interactive communication platform and applied a final online multimodal exam for the fifth-year medical students. This multimodal exam approach used predesigned clinical scenarios that assessed various competencies, including history taking, clinical examination findings, communication skills, laboratory, and diagnostic image interpretation. However, despite adopting this novel approach, it was essential to study the effect of this newly implemented virtual assessment method on both students and faculty. Therefore, we explored the perceptions of the students and faculty toward this newly implemented online multimodal exam, the challenges, limitations, and the overall satisfaction of the participants.

In this study, around 90% of the participants stated that they were familiar and had previous experience with different virtual interactive communication platforms and apps, such as the Zoom™ platform. Over the previous few months, different virtual education platforms had been adopted and implemented by the faculty of medicine and ensured that all students and faculty had unrestricted access to these platforms. This project was adopted by the pediatric department to empower, support, and facilitate distance learning among its staff and students. This went along with the worldwide transition as most academic institutions converted to use online learning platforms as an alternative during this period of pandemic and lockdown to ensure the safety of staff and students [9].

This study demonstrated that a comprehensive multi-format, the high-stakes exam, could be run online uneventfully with an acceptable level of satisfaction by all stakeholders as nearly 84% of participants in the study felt comfortable. Although we expected that students would be anxious about how they would perform in this online exam due to their unfamiliarity with this type of exam, many of them reported that taking the OSCE online reduced their anxiety levels. One possible explanation is that their anxiety toward acquiring COVID-19 during a face-to-face OSCE surpassed their anxiety of being examined online for the first time. This explanation could be supported by the preference of most participants not to continue the remote assessment after the COVID-19 pandemic [10].

However, during an infectious disease outbreak, the literature reported that medical students expressed anxiety during coronavirus disease, with an increased level of social avoidance during the outbreaks [11,12,13]. Previous reports in an academic teaching hospital setting with MERS-CoV experience showed increased knowledge and adherence to protective hygienic practices, and reduced anxiety towards COVID-19 [14].

Our results showed no gender correlations with the perceptions about the e-OSCE experience. Yet, it is important to consider gender-specific should be evaluated in future vitual OSCE, considering the previously reported differences between female and male students [15]. Also, the older age group who perceived lower quality of e-OSCE in our study suggest that more emphasis on their virtual assessment experience may be warranted in the future. Our medical students also perceived the quality of e-OSCE assessment to be significantly worse than the teachers. Kim et al., found that while students were generally prepared for e-learning, there was a significant improvement in their OSCE performance after e-learning interventions [16]. Still, there were gender differences but no age associations; with their readiness being higher for males than females (*p* < 0.05), without differences across ages (*p* = 0.24) [16].

Although participants were overall satisfied with the e-OSCE, they rated some aspects of this new experience more favorably than others. The highly-rated positive aspects of this experience were clear instructions provided prior to the e-OSCE exam and the presence of a team of coordinators, and their prompt responses. Similar findings were observed by Khalaf et al. [17]. Of note, we observed that the participants preferred communication through emails compared to the WhatsApp^®^ group. However, as there were no previous studies with similar findings; this new finding might be due to the perception among the surveyed participants that emails are still being viewed as a more official way of communication in comparison to WhatsApp^®^ [18]. Moreover, some studies have shown that students who had prior exposure to online education were more satisfied with the e-OSCE than those who had not undergone such an experience before. This is understandable and expected as the former are more adept at using online communication platforms than the latter [17,19].

In this study, participants who perceived worse e-OSCE quality were significantly less willing to recommend this virtual assessment in the future. Therefore, more efforts are needed to improve such virtual assessments and mitigate any negative aspects of the experience, for both the students and faculty. Some participants reported minor technical problems as a negative aspect of e-OSCE, such as a slow-speed internet. Similarly, findings were observed by Pal D et al. with the use of Microsoft^®^ teams as an online learning platform during the current COVID-19 pandemic [20]. However, these problems can be overcome with a high-speed internet, better exam coordination, and IT support to provide the required help and support to students who may encounter any technical issues during the exam, as well as clear instructions and guidance given to them prior to the exam.

Although the pediatric e-OSCE was perceived favorably by most students and faculty, it has several limitations. As the COVID-19 pandemic evolved rapidly, the decision to implement the e-OSCE project was taken urgently. Consequently, the sample size was small from a single institution which limits the generalizability of the findings. Information bias is another limitation that cannot be ruled out as well. In addition, this survey was administered to students and faculty in the pediatrics rotation, and their perceptions may differ from students in other clinical rotations. The survey also used closed-ended questions, which may limit the examination of beliefs and attitudes. Despite these limitations, this is the first study that has explored the perceptions of both the students and faculty members about pediatric e-OSCE in Saudi Arabia. Additionally, the positive and negative aspects of this new experience were revealed from both the medical faculty’s and students’ perceptions.

## 5. Conclusions

In this study, a novel way of e-OSCE was implemented to ensure medical education continuity and quality during the COVID-19 crisis. This virtual method of assessment was generally well-perceived by both students and faculty, but older ages were less satisfied with the quality of this virtual assessment. While e-OSCE provides a valuable alternative to the classic face-to-face OSCE during infectious disease outbreaks, more research on improving this virtual assessment tool is warranted. Sharing innovative educational experiences, such as the e-OSCE, in these circumstances is beneficial as the pandemic continues to pose a global threat that could continue to affect medical education and evaluation for a long time. The findings of this study should guide the future direction and decisions toward the optimization of this innovative virtual experiment as well as inform any upcoming studies that are aimed to explore and assess the educational outcomes of different virtual strategies during this unconventional time.

## Figures and Tables

**Figure 1 healthcare-09-00950-f001:**
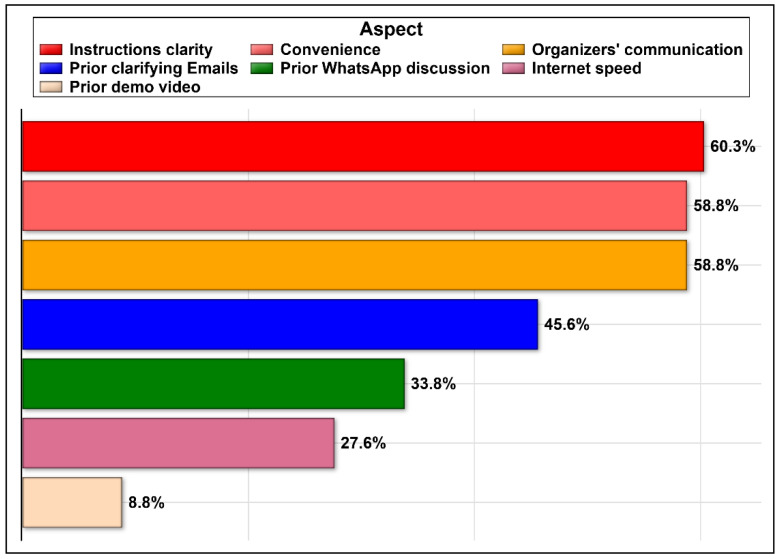
Different aspects of the OSCE experience that were viewed positively by the participants.

**Table 1 healthcare-09-00950-t001:** The e-OSCE survey participating subjects characteristics (*n* = 73).

Characteristic	Frequency	Percentage
**Sex**		
Female	63	86.3
Male	10	13.7
**Age**		
≤30 years	50	68.5
>30 years	23	31.5
**Role**		
Academic teacher	23	31.5
Medical student	50	68.5
**Previously used teleconference methods**		
Face time	26	35.6
Zoom	72	98.6
Webinar	40	54.8
Work-related online meetings	26	35.6
Online learning interfaces	35	47.9
Telephonic conference	15	20.5
Other methods/tools	1	1.4

**Table 2 healthcare-09-00950-t002:** Characteristics of the e-OSCE experience by the faculty and students (*n* = 73).

Question	Frequency	Percentage (%)
**How comfortable did you feel participating in this remote clinical exam (via Zoom or any other similar application)?**		
Not at all comfortable	1	1.4
Not so comfortable	7	9.6
Somewhat comfortable	14	19.2
Very comfortable	35	47.9
Extremely comfortable	16	21.9
**In regard to your previous experience with “classic face-to-face” clinical OSCE, what is the preferred OSCE style for you during the COVID-19 Pandemic?**		
Virtual OSCE (e-OSCE) is preferred	39	53.4
Classic face-to-face is preferred	23	31.5
Both are equally preferred for me	11	15.1
**How do you think these remote clinical exams (e-OSCE) affected the quality of the student’s assessment?**		
Similar assessment to the face-to-face OSCE	34	46.6
Better assessment than face-to-face OSCE	11	15.1
Worse assessment than face-to-face OSCE	28	38.4
**Doing remote video assessment during the COVID-19 pandemic decreased my anxiety-mean (SD) 1–5 Likert agreement.**		
Strongly disagree	6	8.2
Disagree	13	17.8
Neither agree or disagree	18	24.7
Agree	26	35.6
Strongly agree	10	13.7
**Video conferencing as an assessment tool for the pediatric course should be incorporated in next year’s courses.**		
Strongly disagree	5	6.8
Disagree	20	27.4
Neither agree or disagree	19	26
Agree	24	32.9
Strongly agree	5	6.8
**Do you suggest continuing on remote student assessments (via Zoom or similar platforms) after the COVID crisis?**		
Yes	19	26
No	54	74

Abbreviations: OSCE: objective structured clinical examination, e-OSCE: electronic objective structured clinical examination.

**Table 3 healthcare-09-00950-t003:** The e-OSCE teleconference participants perceived “Positive” versus “Negative” aspects about the experience.

Characteristic	Frequency	Percentage (%)
**The participants perceived Positive aspects of the Online OSCE experience**		
The clear instructions provided	41	60.3
Organizers’ prompt communications	40	58.8
The free use of the zoom application	40	58.8
Clarifying emails received from the organizers	31	45.6
The WhatsApp group discussion that related specifically to this event	23	33.8
The fast internet speed	19	27.9
The demo of videoconference before connecting	5	7.4
Other	6	8.8
**The participants perceived negative aspects of the Online OSCE experience**		
Slow internet speed	30	44.1
Unclear demo video	18	26.4
Unclear instructions	10	14.7
Unfamiliarity with the application	6	8.8
Not receiving clarification emails	2	2.9
Other	23	33.8

**Table 4 healthcare-09-00950-t004:** Bivariate analysis of participants’ e-OSCE quality of assessment.

**Variable**	**How Do You Think These Remote Clinical Exams (e-OSCE) Affected the Quality of the Student’s Assessment?**		**Test Statistic**	***p*-Value**
	**Similar/Better**	**Worse**		
Sex				
Female	40 (88.9)	23 (82.1)	χ^2^(1) = 0.22	0.642
Male	5 (11.1)	5 (17.9)		
Age				
≤30 years	35 (77.8)	15 (53.6)	χ^2^(1) = 4.70	0.030
>30 years	10 (22.2)	13 (46.4)		
Role				
Academic teacher/coordinator	10 (22.2)	13 (46.4)	χ^2^(1) = 4.670	0.030
Medical student	35 (77.8)	15 (53.6)		
How likely is it that you would recommend virtual assessment (e-OSCE) to a friend or colleague?-median Likert rating	8	6	U(73) = 380.5	0.003
How comfortable did you feel participating in this remote clinical exam (via Zoom or any other similar application)?—median Likert rating	4	3.4	U(73) = 418	0.010
In regard to your previous experience with “classic face-to-face” clinical OSCE, what is the preferred OSCE style for you during the COVID Pandemic?				
virtual OSCE (e-OSCE) is preferred	29 (64.4)	10 (35.7)	χ^2^(2) = 7.60	0.022
classic face-to-face is preferred	9 (20)	14 (50)		
Both are equally preferred for me	7 (15.6)	4 (14.3)		
Doing remote video assessment during the COVID-19 pandemic decreased my anxiety-median Likert agreement	4	3	U(73) = 487	0.093
Video conferencing as an assessment tool for the pediatric course should be incorporated in next year’s courses-median value	4	2	U(73) = 223.5	<0.001

## Data Availability

The data are available upon reasonable request from the corresponding author.

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
