# Peer review of "Medical Faculty’s and Students’ Perceptions toward Pediatric Electronic OSCE during the COVID-19 Pandemic in Saudi Arabia"

_healthcare, 2021, doi:10.3390/healthcare9080950_

Round 1
Reviewer 1 Report
This is a cross-sectional, ex post facto online survey of a convenience sample of professors and students regarding their perceptions of an electronic OSCE conducted at one medical school in Saudi Arabia. There are a number of concerns that may be barriers to publication.
General: Ethics review is usually included in the methods section. There is no description of the educational content that was tested. These deficiencies must be corrected, if possible.
Title: descriptive of the study
Abstract: pertinent findings are described.
Key words: most of the key words are not MeSH terms. Omit 'transition', 'electronic OSCE', 'pediatric assessment', and 'final year.' Add 'distance learning', 'assessment, educational', and 'pediatric.'
A description of the pediatric case and testable objectives would strengthen your manuscript. An supplemental file could be used for this purpose. It would also be interesting to know what components of the clinical encounter were relatively easy for the student to conduct from a distance and which ones were more difficult or not possible. For example, were live pediatric patients used? Were additional electronic devices (blood pressure cuffs, point-of-care devices, etc. attached to computers) used to collect biometric data from patients? How was physical assessment conducted and measured? In what ways was the face-to-face v. virtual encounter different? How did the faculty modify the face-to-face so that it would "work" from a distance?
When was the data collected relative to the OSCE? The time frame in which the online survey was available should be noted in the methods.
Limitations are well described in the discussion section.
References are not in mdpi style.
Thank you for the opportunity to review your manuscript.
Author Response
Reviewer 1:
Comments and Suggestions for Authors
This is a cross-sectional, ex post facto online survey of a convenience sample of professors and students regarding their perceptions of an electronic OSCE conducted at one medical school in Saudi Arabia. There are a number of concerns that may be barriers to publication.
Reply: Thanks for review. We have addressed your valuable suggestions in the revised manuscript, as per the followings points below.
General: Ethics review is usually included in the methods section. There is no description of the educational content that was tested. These deficiencies must be corrected, if possible.
Reply:
Thank you. For the Ethical considerations: the last paragraph in the Methods section stated “Those who accepted the invitation were asked to consent to participate before filling out the questionnaire. No personal identifiers were collected, and the study adhered to the ethical principles of the Helsinki’s declaration.”
As for the no description of the educational content that was tested: The objectives of the exam were to test the student’s ability to take a comprehensive history including the different unique components in common pediatric cases. The e-OSCE stations were similar in themes and structure to the traditional OSCE that were conducted face-to-face before the COVID-19 crisis.
The general themes of the stations included: taking a developmental history and estimating the child’s developmental age, asthma station with questions in acute management and long-term management, bronchiolitis station. Also included was a febrile seizure station and the questions included history taking, differential diagnosis and distinguishing between febrile seizure and other types of seizures. In addition, a station included interpretation of different patterns of the growth chart. A station was targeted at identifying different equipment in pediatrics and the ability to describe the indications, contradictions, and adverse events.
Furthermore, case stations included abdominal pain and abdominal mass as well as a neonatology or critical care station were also included. Each station’s allocated time was 8 minutes per station. Each student had five stations to go through. Each station of the OSCE had 3-4 questions with 2 minutes time allocated to answer each question.
We included brief description of the e-OSCE in the revised paper.
Title: descriptive of the study
Reply: Thanks for the reviewer’s comment.
Abstract: pertinent findings are described.
Reply: Thanks for the reviewer’s comment.
Key words: most of the key words are not MeSH terms. Omit 'transition', 'electronic OSCE', 'pediatric assessment', and 'final year.' Add 'distance learning', 'assessment, educational', and 'pediatric.'
Reply: As per your valuable comment the keywords were changed to: ‘distance learning', 'assessment, educational', and 'pediatric.' in the revised version.
A description of the pediatric case and testable objectives would strengthen your manuscript. An supplemental file could be used for this purpose. It would also be interesting to know what components of the clinical encounter were relatively easy for the student to conduct from a distance and which ones were more difficult or not possible. For example, were live pediatric patients used? Were additional electronic devices (blood pressure cuffs, point-of-care devices, etc. attached to computers) used to collect biometric data from patients? How was physical assessment conducted and measured? In what ways was the face-to-face v. virtual encounter different? How did the faculty modify the face-to-face so that it would "work" from a distance?
When was the data collected relative to the OSCE? The time frame in which the online survey was available should be noted in the methods.
Reply:
In regards to the testable material, as we stated above:
The objectives of the exam were to test the student’s ability to take a comprehensive history including the different unique components in common pediatric cases. The e-OSCE stations were similar in themes and structure to the traditional OSCE that were conducted face-to-face before the COVID-19 crisis.
The general themes of the stations included: taking a developmental history and estimating the child’s developmental age, asthma station with questions in acute management and long-term management, bronchiolitis station. Also included was a febrile seizure station and the questions included history taking, differential diagnosis and distinguishing between febrile seizure and other types of seizures. In addition, a station included interpretation of different patterns of the growth chart. A station was targeted at identifying different equipment in pediatrics and the ability to describe the indications, contradictions, and adverse events.
Furthermore, case stations included abdominal pain and abdominal mass as well as a neonatology or critical care station were also included. Each station’s allocated time was 8 minutes per station. Each student had five stations to go through. Each station of the OSCE had 3-4 questions with 2 minutes time allocated to answer each question.
We included brief description of the e-OSCE in the revised paper.
Limitations are well described in the discussion section.
Reply:
Thank you. The last paragraph in the Discussion was elaborated on the study limitations:
“Although the paediatric e-OSCE was perceived favourably by most students and faculty, it has several limitations. As the COVID-19 pandemic evolved rapidly, the decision to implement the e-OSCE project was taken urgently. Consequently, the sample size was small from a single institution which limits the generalizability of the findings. In-formation bias is another limitation that cannot be ruled out as well. Also, this survey was administered to students and faculty in the pediatrics rotation and their perceptions may differ from students in other clinical rotations. In addition, the survey used closed-ended questions which may limit the examination of beliefs and attitudes.”
References are not in mdpi style.
Reply: Thank you the references were edited in the Revised Manuscript in the MDPI style.
Thank you for the opportunity to review your manuscript.

Reviewer 2 Report
The article reveals an experience in the context of emergency remote education and in the processes of learning assessment within health professionals training. The subject of this paper is of increasing interest. It reveals an alternative to face-to-face classes that are not held for pandemic reasons. It was clear that online teaching has advantages and disadvantages that are interesting to know and reflect on, in order to improve the teaching-learning processes.
However, it is suggested to the authors:
- Perhaps, reading some scholarly articles will help you to rearranged and complete the Introdution section. Please, add more references;
- That clearly explain the objectives of the investigation or define hypotheses to be studied;
- That the relationship of the objectives defined for the investigation with the results and conclusions presented is evident. It is fundamentally to make explicit the internal coherence, expressed in the relationship that has just been presented;
- Unfortunately, the resulting findings appear insufficient to the research. Present some results derived from the statistical analysis with SPSS, in addition to the descriptive analysis of frequencies. It suggests the search for dependence relations among the variables under study;
- Authors should indicate study limitations.
Author Response
Reviewer 2:
Comments and Suggestions for Authors
The article reveals an experience in the context of emergency remote education and in the processes of learning assessment within health professionals training. The subject of this paper is of increasing interest. It reveals an alternative to face-to-face classes that are not held for pandemic reasons. It was clear that online teaching has advantages and disadvantages that are interesting to know and reflect on, in order to improve the teaching-learning processes.
Reply: Thanks for review. We have addressed your valuable suggestions in the revised manuscript, as per the followings.
However, it is suggested to the authors:
- Perhaps, reading some scholarly articles will help you to rearranged and complete the Introdution section. Please, add more references
Reply:
Thanks for your valuable comments, we rearranged the introduction and included additional relevant references in the revised paper.
- That clearly explain the objectives of the investigation or define hypotheses to be studied;
Reply:
Thanks for your comment. In this study, our objective is to examine the perceptions of students and faculty on the implementation of e-OSCE during COVID-19 lockdown in King Saud University in Riyadh, Saudi Arabia. This has been reflected in the revised manuscript.
- That the relationship of the objectives defined for the investigation with the results and conclusions presented is evident. It is fundamentally to make explicit the internal coherence, expressed in the relationship that has just been presented.
Reply:
Thank you for your comment. The objective of this study was clearly defined to explore the perceptions of the faculty and students about the newly implemented e-OSCE as compared to the traditional face-to-face OSCE, and whether e-OSCE is as effective as the traditional model (from the participants’ experience). This is also reflected in the results section.
- Unfortunately, the resulting findings appear insufficient to the research. Present some results derived from the statistical analysis with SPSS, in addition to the descriptive analysis of frequencies. It suggests the search for dependence relations among the variables under study;
Reply:
Thanks for your valuable comment, we added Table 4 and the following comparisons to the Results:
“To understand better what may explain why participants had perceived different quality of e-OSCE assessment, the bivariate analysis was conducted (Table 4). The resulted findings showed that the student’s sex did not correlate significantly on their perception of the quality of e-OSCE assessment, but ages older than thirty years perceived the quality of e-OSCE assessment significantly worse (p=0.030). Also, medical students perceived the quality of e-OSCE assessment significantly worse as compared to teachers (p=0.030) according to the chi-squared test of independence. The non-parametric Mann-Whitney U test showed that physicians who perceived worse e-OSCE quality were significantly less willing to recommend this virtual assessment (Median=6) as compared to those who perceived it as better or similar to the face-face assessment (Median=8) on average (U=350.5, df=73, p=0.003).
Another non-parametric Mann-Whitney U test showed that the students who perceived worse e-OSCE quality had perceived significantly lower comfort (Median= 3.4) as compared to those who had perceived the e-OSCE equivalently or better with respect to the quality of assessment compared to the face-face methods (U=418, df=73, p=0.010). Even so, participants who preferred the face-face OSCE assessment were found to be significantly more predicted to perceive the e-OSCE as significantly worse than face-face methods (p=0.022). People’s perception of anxiety during the e-OSCE did not correlate significantly with their perceived quality of the virtual assessment (p=0.093), but those who perceived the e-OSCE as worse had measured significantly lower willingness to incorporate virtual assessments in future courses ( median=2) compared to people who were satisfied with the quality of the e-OSCE method ( median=4) , according to a Mann-Whitney U test (p<0.001).”
- Authors should indicate study limitations.
Reply:
Thanks for your valuable comment. The last paragraph in the Discussion was elaborated on the study limitations:
“Although the paediatric e-OSCE was perceived favourably by most students and faculty, it has several limitations. As the COVID-19 pandemic evolved rapidly, the decision to implement the e-OSCE project was taken urgently. Consequently, the sample size was small from a single institution which limits the generalizability of the findings. In-formation bias is another limitation that cannot be ruled out as well. Also, this survey was administered to students and faculty in the pediatrics rotation and their perceptions may differ from students in other clinical rotations. In addition, the survey used closed-ended questions which may limit the examination of beliefs and attitudes.”

Round 2
Reviewer 2 Report
With the new Results, the Discussion and Conclusion section must be completed and improved.
Author Response
- Thanks for your valuable comments.
- We have improved further the Introduction part.
- We also elaborated more on the Discussion and Conclusion section as per the added results as per the valuable suggestion:
Discussion (Paragraph added:
Our results showed no gender correlations with the perceptions about the e-OSCE experience. Yet, it is important to consider gender-specific should be evaluated in future vitual OSCE, considering the previously reported differences between female and male students [15]. Also, older age group who perceived lower quality of e-OSCE in our study suggest that more emphasis on their virtual assessment experience may be warranted in the future. Our medical students also perceived the quality of e-OSCE assessment significantly worse than teachers. Kim et al. found that while students were generally prepared for e-learning, there was a significant improvement in their OSCE performance after e-learning interventions[16]. Still, there were gender differences but no age associations; with their readiness being higher for males than females (p < 0.05), without differences across ages (p = 0.24) [16].
...
In this study, participants who perceived worse e-OSCE quality were significantly less willing to recommend this virtual assessment in the future. Therefore, more efforts are needed to improve such virtual assessments and mitigate any negative aspects of the experience, for both the students and faculty. Some participants reported minor technical ...
Conclusion:
In this study, a novel way of e-OSCE was implemented to ensure medical education continuity and quality during the COVID-19 crisis. This virtual method of assessment was generally well-perceived by both students and faculty, but older ages were less satisfied with the quality of this virtual assessment. While e-OSCE provides a valuable alternative to the classic face-to-face OSCE during infectious disease outbreaks, more research on improving this virtual assessment tool is warranted. Sharing innovative educational experiences, such as the e-OSCE, in these circumstances is beneficial as the pandemic continues to pose a global threat that could continue to affect medical education and evaluation for a long time. The findings of this study should guide the future direction and decisions toward the optimization of this innovative virtual experiment and inform upcoming studies that are aimed to explore and assess the educational outcomes of different virtual strategies during this unconventional time.
- The English language and style underwent fine/minor spell check as advised